# Bayesian Structure Learning by Recursive Bootstrap

**Raanan Y. Rohekar**[*]
Intel AI Lab
raanan.yehezkel@intel.com

**Yaniv Gurwicz**[*]
Intel AI Lab
yaniv.gurwicz@intel.com

**Shami Nisimov**[*]
Intel AI Lab
shami.nisimov@intel.com

**Guy Koren**
Intel AI Lab
guy.koren@intel.com

**Gal Novik**
Intel AI Lab
gal.novik@intel.com

## Abstract

We address the problem of Bayesian structure learning for domains with hundreds of variables by employing non-parametric bootstrap, recursively. We propose a method that covers both model averaging and model selection in the same framework. The proposed method deals with the main weakness of constraint-based learning—sensitivity to errors in the independence tests—by a novel way of combining bootstrap with constraint-based learning. Essentially, we provide an algorithm for learning a tree, in which each node represents a scored CPDAG for a subset of variables and the level of the node corresponds to the maximal order of conditional independencies that are encoded in the graph. As higher order independencies are tested in deeper recursive calls, they benefit from more bootstrap samples, and therefore are more resistant to the curse-of-dimensionality. Moreover, the re-use of stable low order independencies allows greater computational efficiency. We also provide an algorithm for sampling CPDAGs efficiently from their posterior given the learned tree. That is, not from the full posterior, but from a reduced space of CPDAGs encoded in the learned tree. We empirically demonstrate that the proposed algorithm scales well to hundreds of variables, and learns better MAP models and more reliable causal relationships between variables, than other state-of-the-art-methods.

## 1  Introduction

Bayesian networks (BN) are probabilistic graphical models, commonly used for probabilistic inference, density estimation, and causal modeling (Darwiche, 2009; Pearl, 2009; Murphy, 2012; Spirtes et al., 2000). The graph of a BN is a DAG over random variables, encoding conditional independence assertions. Learning this DAG structure, $G$, from data, $D$, has been a fundamental problem for the past two decades. Often, it is desired to learn an equivalence class (EC) of DAGs, that is, a CPDAG. DAGs in an EC are Markov equivalent; that is, given an observed dataset, they are statistically indistinguishable and represent the same set of independence assertions (Verma & Pearl, 1990).

Commonly, two main scenarios are considered. In one scenario, the posterior probability, $P(G|D)$, (or some other structure scoring metric) peaks sharply around a single structure, $G_{\mathrm{MAP}}$. Here, the goal is to find the highest-scoring structure—a maximum-a-posteriori (MAP) estimation (model selection). In a second scenario, several distinct structures have high posterior probabilities, which is common when the data size is small compared to the domain size (Friedman & Koller, 2003). In this case, learning model structure or causal relationships between variables using a single MAP model may give unreliable conclusions. Thus, instead of learning a single structure, graphs are sampled

---

[*]Equal contribution.

from the posterior probability, $P(G|D)$ and the posterior probabilities of hypotheses-of-interests, e.g., structural features, $f$, are computed in a model averaging manner. Examples of structural features are: the existence of a directed edge from node $X$ to node $Y$, $X \to Y$, a Markov blanket feature, $X \in \mathrm{MB}(Y)$, and a directed path feature $X \rightsquigarrow Y$. Another example is the computation of the posterior predictive probability, $P(D^{\mathrm{new}}|D)$. In the model selection scenario, it is equal to $P(D^{\mathrm{new}}|G_{\mathrm{MAP}})$. In the model averaging scenario, it is equal to averaging over all the DAG structures, $\mathcal{G}$. That is, $\sum_{G \in \mathcal{G}} P(D^{\mathrm{new}}|G)P(G|D)$.

The number of DAG structures is super exponential with the number of nodes, $O(n!2^{\binom{n}{2}})$, rendering an exhaustive search for an optimal DAG or averaging over all the DAGs intractable for many real-world problems. In fact, it was shown that recovering an optimal DAG with a bounded in-degree is NP-hard (Chickering et al., 1995).

In this paper we propose: (1) an algorithm, called B-RAI, that learns a generative tree, $\mathcal{T}$, for CPDAGs (equivalence classes), and (2) an efficient algorithm for sampling CPDAGs from this tree. The proposed algorithm, B-RAI, applies non-parametric bootstrap in a recursive manner, and combines CI-tests and scoring.

## 2 Related Work

Previously, two main approaches for structure learning were studied, score-based (search-and-score) and constraint-based. Score-based approaches combine a scoring function, such as BDe (Cooper & Herskovits, 1992), with a strategy for searching through the space of structures, such as greedy equivalence search (Chickering, 2002). Constraint-based approaches (Pearl, 2009; Spirtes et al., 2000) find the optimal structures in the large sample limit by testing conditional independence (CI) between pairs of variables. They are generally faster than score-based approaches, scale well for large domains, and have a well-defined stopping criterion (e.g., maximal order of conditional independence). However, these methods are sensitive to errors in the independence tests, especially in the case of high-order conditional-independence tests and small training sets. Some methods are a hybrid between the score-based and constraint-based methods and have been empirically shown to have superior performance (Tsamardinos et al., 2006).

Recently, important advances have been reported for finding optimal solutions. Firstly, the efficiency in finding an optimal structure (MAP) has been significantly improved (Koivisto & Sood, 2004; Silander & Myllymäki, 2006; Jaakkola et al., 2010; Yuan & Malone, 2013). Secondly, several methods for finding the $k$-most likely structures have been proposed (Tian et al., 2010; Chen & Tian, 2014; Chen et al., 2016). Many of these advances are based on defining new search spaces and efficient search strategies for these spaces. Nevertheless, they are still limited to relatively small domains (up to 25 variables). Another type of methods are based on MCMC (Friedman & Koller, 2003; Eaton & Murphy, 2007; Grzegorczyk & Husmeier, 2008; Niinimäki & Koivisto, 2013; Su & Borsuk, 2016) where graphs are sampled from the posterior distribution. However, there is no guarantee on the quality of the approximation in finite runs (may not mix well and converge in finite runs). Moreover, these methods have high computational costs, and, in practice, they are restricted to small domains.

## 3 Proposed Method

We propose learning a tree, $\mathcal{T}$, by applying non-parametric bootstrap recursively, testing conditional independence, and scoring the leaves of $\mathcal{T}$ using a Bayesian score.

### 3.1 Recursive Autonomy Identification

We first briefly describe the RAI algorithm, proposed by Yehezkel & Lerner (2009), which given a dataset, $\mathcal{D}$, constructs a CPDAG in a recursive manner. RAI is a constraint-based structure learning algorithm. That is, it learns a structure by performing independence tests between pairs of variables conditioned on a set of variables (CI-tests). As illustrated in Figure 1, the CPDAG is constructed recursively, from level $n = 0$. In each level of recursion, the current CPDAG is firstly refined by removing edges between nodes that are independent conditioned on a set of size $n$ and directing the edges. Then, the CPDAG is partitioned into ancestors, $\boldsymbol{X}_{\mathrm{A}i}^{(n)}$, and (2) descendant, $\boldsymbol{X}_{\mathrm{D}}^{(n)}$ groups.

Each group is *autonomous* in that it includes the parents of its members (Yehezkel & Lerner, 2009). Further, each autonomous group from the $n$-th recursion level, is independently partitioned, resulting in a new level of $n + 1$. Each such CPDAG (a subgraph over the autonomous set) is progressively partitioned (in a recursive manner) until a termination condition is satisfied (independence tests with condition set size $n$ cannot be performed), at which point the resulting CPDAG (a subgraph) at that level is returned to its parent (the previous recursive call). Similarly, each group in its turn, at each recursion level, gathers back the CPDAGs (subgraphs) from the recursion level that followed it, and then return itself to the recursion level that precedes it, and until the highest recursion level, $n = 0$, is reached, and the final CPDAG is fully constructed.

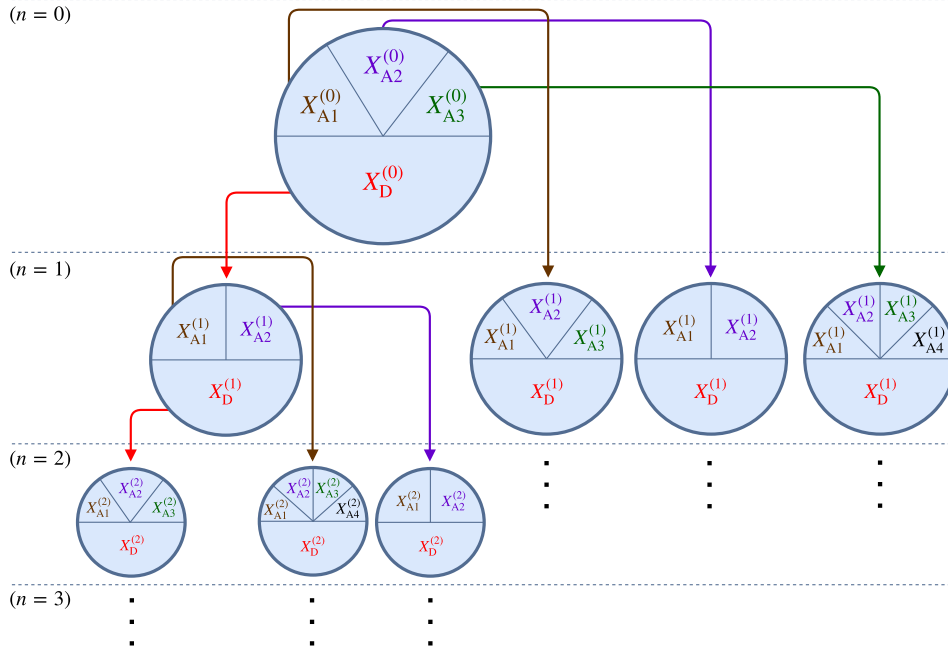

Figure 1: An example of an execution tree of RAI. An arrow indicates a recursive call. Each CPDAG is partitioned into ancestors group, $\boldsymbol{X}_{Ai}^{(n)}$, and descendant group, $\boldsymbol{X}_{D}^{(n)}$, and then, each group is further partitioned, recursively with $n + 1$. Each circle represents a distinct subset of variables (for example, $\boldsymbol{X}_{A_1}^{(1)}$ in different circles represents different subsets). Best viewed in color.

## 3.2 Uncertainty in Conditional Independence Test

Constraint-based structure learning algorithms, such as RAI, are proved to recover the true underlying CPDAG when using an optimal independence test. In practice, independence is estimated from finite-size, noisy, datasets. For example, the dependency between $X$ and $Y$ conditioned on a set $\boldsymbol{Z} = \{Z_i\}_i^l$ is estimated by thresholding the conditional mutual information,

$$\widehat{\mathrm{MI}}(X, Y | \boldsymbol{Z}) = \sum_X \sum_Y \sum_{Z_1} \cdots \sum_{Z_l} P(X, Y, \boldsymbol{Z}) \log \frac{P(X, Y | \boldsymbol{Z})}{P(X | \boldsymbol{Z}) P(Y | \boldsymbol{Z})}, \tag{1}$$

where the probabilities are estimated from a limited dataset $\mathcal{D}$. Obviously, this measure suffers from the curse-of-dimensionality, where for large condition set sizes, $l$, this measure becomes unreliable.

The relation between the optimal conditional mutual information, $\mathrm{MI}$, and the conditional mutual information, $\widehat{\mathrm{MI}}$, estimated from a limited dataset $\mathcal{D}$, is

$$\widehat{\mathrm{MI}}(X, Y | \boldsymbol{Z}) = \mathrm{MI}(X, Y | \boldsymbol{Z}) + \sum_{m=1}^{\infty} C_m + \epsilon, \tag{2}$$

where $\sum_{m=1}^{\infty} C_m$ is an estimate of the average bias for limited data (Treves & Panzeri, 1995), and $\epsilon$ is a zero mean random variable with unknown distribution. Lerner et al. (2013) proposed thresholding

$\widehat{\mathrm{MI}}$ with the leading term of the bias, $C_1$, to test independence. Nevertheless, there is still uncertainty in the estimation due to the unknown distribution of $\epsilon$, which may lead to erroneous independence assertions. One inherent limitation of the RAI algorithm, as well as other constraint-based algorithms, is its sensitivity to errors in independence testing. An error in an early stage, may lead to additional errors in later stages. We propose modeling this uncertainty using non-parametric bootstrap.

The bootstrap principle is to approximate a population distribution by a sample distribution (Efron & Tibshirani, 1994). In its most common form, the bootstrap takes as input a data set $\mathcal{D}$ and an estimator $\psi$. To generate a sample from the bootstrapped distribution, a dataset $\widetilde{\mathcal{D}}$ of cardinality equal to that of $\mathcal{D}$ is sampled uniformly with replacement from $\mathcal{D}$. The bootstrap sample estimate is then taken to be $\psi(\widetilde{\mathcal{D}})$. When this process is repeated several times, it produces several resampled datasets, estimators and thereafter sample estimates, from which a final estimate can be made by MAP or model averaging (Friedman et al., 1999). The bootstrap is widely acclaimed as a great advance in applied statistics and even comes with theoretical guarantees (Bickel & Freedman, 1981).

We propose estimating the result of the $n+1$ recursive call ($\psi$) for each autonomous group using non-parametric bootstrap.

### 3.3 Graph Generative Tree

We now describe a method for constructing a tree, $\mathcal{T}$, from which CPDAGs can be sampled. In essence, we replace each node in the execution tree, as illustrated in Figure 1, with a *bootstrap-node*, as illustrated in Figure 2. In the bootstrap-node, for each autonomous group ($\boldsymbol{X}_{\mathrm{A}i}^{(n)}$ and $\boldsymbol{X}_{\mathrm{D}}^{(n)}$), $s$ datasets, $\{\widetilde{\mathcal{D}}_t\}_{t=1}^s$, are sampled with replacement from the training data $\mathcal{D}$, where $|\widetilde{\mathcal{D}}_t| = |\mathcal{D}|$. This results in a recursive application of bootstrap. Finally, we calculate $\log[P(D|G)]$ for each leaf node in the tree ($G$ is the CPDAG in the leaf), using a decomposable score,

$$\log[P(\mathcal{D}|G)] = \sum_i \mathrm{score}(X_i|\pi_i; \mathcal{D}), \tag{3}$$

where $\pi_i$ are the parents of node $X_i$. For example, Bayesian score (Heckerman et al., 1995).

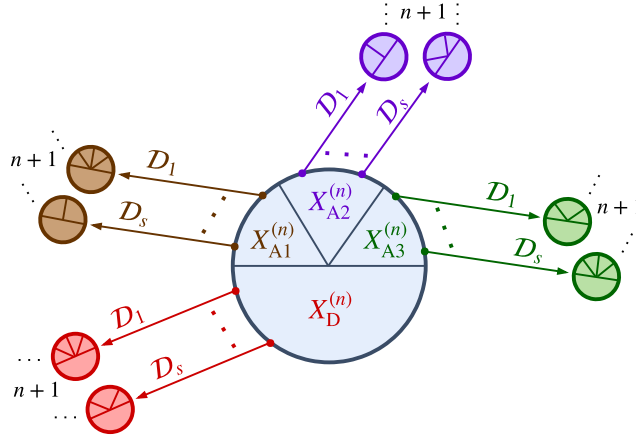

Figure 2: The bootstrap node. For each autonomous group, $s$ recursive calls are performed. Each call uses a sampled dataset, $\widetilde{\mathcal{D}}_t \sim \mathcal{D}$ where $t \in \{1, \cdots, s\}$. Best viewed in color.

The recursive construction of $\mathcal{T}$ is described in Algorithm 1. The algorithm starts with condition set size $n = 0$, $G$ a complete graph, and a set of exogenous variables $X_{\mathrm{ex}} = \emptyset$. The set $X_{\mathrm{ex}}$ is exogenous to $G$ and consists the parents of $\boldsymbol{X}$. First, an exit condition is tested (line 2). It is satisfied if there are not enough variables for a condition set of size $n$. In this case, $\mathcal{T}$ is set to be a leaf node, and the input graph $G$ is scored using the full training data (not the sampled data). It is important to note that only leaf nodes of $\mathcal{T}$ are scored. From this point, the recursive procedure will trace back, adding parent nodes to $\mathcal{T}$.

---

**Algorithm 1:** Construct a graph generative tree, $\mathcal{T}$

---

1  $\mathcal{R} \longleftarrow$ **B-RAI** $(G, \boldsymbol{X}, \boldsymbol{X}_{\mathrm{ex}}, n, \mathcal{D}, \widetilde{\mathcal{D}})$

    **Input:** an initial CPDAG $G$ over endogeneous $\boldsymbol{X}$ & exogenous nodes $\boldsymbol{X}_{\mathrm{ex}}$, a desired resolution $n$, full
        training data $\mathcal{D}$, and sampled training data $\widetilde{\mathcal{D}}$.

    **Output:** $\mathcal{R}$, root of the graph generative tree $\mathcal{T}$.

2    **if** $\max_{X_i \in \boldsymbol{X}}(|\pi_i| - 1) < n$ **then**          ▷ exit condition (test maximal indegree)

3       $sc \longleftarrow \texttt{Score}(G, \mathcal{D})$

4       $\mathcal{R} \longleftarrow$ a leaf node with content $(G, sc)$

5       **return** $\mathcal{T}$

6    $G' \longleftarrow \texttt{IncreaseResolution}(G, n, \widetilde{\mathcal{D}})$          ▷ $n$-th order independencies

7    $\{\boldsymbol{X}_{\mathrm{D}}, \boldsymbol{X}_{\mathrm{A}1}, \ldots, \boldsymbol{X}_{\mathrm{A}K}\} \longleftarrow \texttt{SplitAutonomous}(\boldsymbol{X}, G')$      ▷ identify autonomies

8    $\mathcal{R} \longleftarrow$ a new root node                      ▷ a bootstrap node

9    **for** $t \in 1 \ldots s$ **do**

10      $\widetilde{\mathcal{D}}' \longleftarrow$ sample with replacement from $\mathcal{D}$          ▷ bootstrap

11      **for** $i \in 1 \ldots K$ **do**

12        $\mathcal{R}_{\mathrm{A}i}^{t} \longleftarrow$ B-RAI$(G', \boldsymbol{X}_{\mathrm{A}i}, \boldsymbol{X}_{\mathrm{ex}}, n+1, \mathcal{D}, \widetilde{\mathcal{D}}')$    ▷ a recursive call

13        set $\mathcal{R}_{\mathrm{A}i}^{t}$ to be the child of $\mathcal{R}$ with label: $\mathrm{Anc}_i^t$

14      $\mathcal{R}_{\mathrm{D}}^{t} \longleftarrow$ B-RAI$(G', \boldsymbol{X}_{\mathrm{D}}, \boldsymbol{X}_{\mathrm{ex}} \cup \{\boldsymbol{X}_{\mathrm{A}i}\}_{i=1}^{K}, n+1, \mathcal{D}, \widetilde{\mathcal{D}}')$    ▷ a recursive call

15      set $\mathcal{R}_{\mathrm{D}}^{t}$ to be the child of $\mathcal{R}$ with label: $\mathrm{Dec}^t$

16    **return** $\mathcal{R}$

---

The procedure $\texttt{IncreaseResolution}$ (line 6) disconnects conditionally independent variables in two steps. First, it tests dependency between $\boldsymbol{X}_{\mathrm{ex}}$ and $\boldsymbol{X}$, i.e., $X \perp\!\!\!\perp X' | \boldsymbol{S}$ for every connected pair $X \in \boldsymbol{X}$ and $X' \in \boldsymbol{X}_{\mathrm{ex}}$ given a condition set $\boldsymbol{S} \subset \{\boldsymbol{X}_{\mathrm{ex}} \cup \boldsymbol{X}\}$ of size $n$. Next, it tests dependencies within $\boldsymbol{X}$, i.e., $X_i \perp\!\!\!\perp X_j | \boldsymbol{S}$ for every connected pair, $X_i, X_j \in \boldsymbol{X}$, given a condition set $\boldsymbol{S} \subset \{\boldsymbol{X}_{\mathrm{ex}} \cup \boldsymbol{X}\}$ of size $n$. After removing the corresponding edges, the remaining edges are directed by applying two rules (Pearl, 2009; Spirtes et al., 2000). First, v-structures are identified and directed. Then, edges are continually directed, by avoiding the creation of new v-structures and directed cycles, until no more edges can be directed. Following the terminology of Yehezkel & Lerner (2009), we say that $G'$ is set by increasing the graph d-separation resolution from $n-1$ to $n$.

The procedure $\texttt{SplitAutonomous}$ (line 7) identifies autonomous sets, one descendant set, $\boldsymbol{X}_{\mathrm{D}}$, and $K$ ancestor sets, $\boldsymbol{X}_{\mathrm{A}1}, \ldots, \boldsymbol{X}_{\mathrm{A}K}$ in two steps. First, the variables having the lowest topological order (the highest indexes in a topological sort) are grouped into $\boldsymbol{X}_{\mathrm{D}}$. Specifically, $\boldsymbol{X}_{\mathrm{D}}$ consists of all the nodes without outgoing directed edges (undirected edges between them may be present). Then, $\boldsymbol{X}_{\mathrm{D}}$ is removed (temporarily) from $G'$ revealing unconnected sub-structures. The number of unconnected sub-structures is denoted by $K$ and the nodes set of each sub-structure is denoted by $\boldsymbol{X}_{\mathrm{A}i}$ $(i \in \{1 \ldots K\})$.

An autonomous set in $G'$ includes all its nodes' parents (complying with the Markov property) and therefore a sub-tree can further be constructed independently, using a recursive call with $n+1$. First, $s$ datasets are sampled from $\mathcal{D}$ (line 10) and the algorithm is called recursively for each dataset and for each autonomous set (for ancestor sets in line 12, and descendant set in line 14). This recursive decomposition of $\boldsymbol{X}$ is similar to that of RAI (Figure 1). The result of each recursive call is a tree. These trees are merged into a single tree, $\mathcal{T}$, by setting a common parent node, $\mathcal{R}$ (line 8), for the roots of each subtree (line 15). From the resulting tree, CPDAGs can be generated (sampled), as described in the next section. Thus, we call it a *graph generative tree* (GGT).

**Complexity**. The computational complexity of two main operations are analyzed, CI tests and scoring. The number of CI tests performed by B-RAI has a complexity of $\mathcal{O}(n^k s^{k+1})$, where $s$ is the number of splits, $n$ is the number of variables, and $k$ is the maximal order of conditional independence in the data. The running-trace of RAI in the worst-case scenario can be viewed as a single path in the GGT (root to leaf). This has a complexity of $\mathcal{O}(n^k)$. Thus, for the number of CI

tests, the ratio between B-RAI and RAI is $\sum_{i=0}^{k} s^i$. For the Bayesian scoring function (scoring a node given its parents), the complexity is $\mathcal{O}(ns^k)$, as only the leaves of the GGT are scored. Note that the worst-case scenario is the case where the true underlying graph is a complete graph, which is not typical in real-world cases. In practice, significantly fewer CI tests and scoring operations are performed, as evident in the short run-times in our experiments.

### 3.3.1 Sampling CPDAGs

In essence, following a path along the learned GGT, $\mathcal{T}$, following one of $s$ possible labeled children at each bootstrap node, results in a single CPDAG. In Algorithm 2 we provide a method for sampling CPDAGs proportionally to their scores. The scores calculation and CPDAGs selections from the GGT are performed backwards, from the leaves to the root (as opposed to the tree construction which is performed in top down manner). For each autonomous group, given $s$ sampled CPDAGs and their scores returned from $s$ recursive calls (lines 8 & 13), the algorithm samples one of the $s$ results (lines 9 & 14) proportionally to their (log) score. We use the Boltzmann distribution,

$$P(t; \{sc^{t'}\}_{t'=1}^{s}) = \frac{\exp[sc^t/\gamma]}{\sum_{t'=1}^{s} \exp[sc^{t'}/\gamma]}, \tag{4}$$

where $\gamma$ is a "temperature" term. When $\gamma \to \infty$, results are sampled from a uniform distribution, and when $\gamma \to 0$ the index of the maximal value is selected ($\arg\max$). We set $\gamma = 1$ and use the Bayesian score, BDeu (Heckerman et al., 1995). Finally, the sampled CPDAGs are merged (line 16) and the sum of scores of all autonomous sets (line 17) is the score of the merged CPDAG.

Another common task is finding the CPDAG having the highest score (model selection). In our case, $G_{\text{MAP}} = \arg\max_{G \in \mathcal{T}} [\log P(\mathcal{D}|G)]$. The use of a decomposable score enables an efficient recursive algorithm to recover $G_{\text{MAP}}$. Thus, this algorithm is similar to Algorithm 2, where sampling (lines 9 & 14) is replaced by $t' = \arg\max_t P(t; \{sc^t\}_{t=1}^{s})$.

---

**Algorithm 2:** Sample a CPDAG from $\mathcal{T}$

---

1   $(G, sc) \longleftarrow$ **SampleCPDAG** $(\mathcal{R})$

    **Input:** $\mathcal{R}$, root of a graph generative tree, $\mathcal{T}$
    **Output:** $G$, a sampled CPDAG and score $sc$

2    **if** $\mathcal{R}$ *is a leaf node* **then**                                   ▷ exit condition
3      $(G, sc) \longleftarrow$ content of the leaf node $\mathcal{R}$
4      **return** $(G, sc)$

5    **for** $i \in 1 \dots K$ **do**
6      **for** $t \in 1 \dots s$ **do**
7        $\mathcal{R}' \longleftarrow \text{Child}(\mathcal{R}, \text{Anc}_i^t)$                ▷ select $\text{Anc}_i$ sub-tree
8        $(G^t, sc^t) \longleftarrow \text{SampleCPDAG}(\mathcal{R}')$        ▷ a recursive call

9      sample $t' \sim P(t'; \{sc^t\}_{t=1}^{s})$ (see Equation 4)
10     $G_{\text{A}i} \longleftarrow G^{t'}$ and $sc_{\text{A}i} \longleftarrow sc^{t'}$

11   **for** $t \in 1 \dots s$ **do**
12     $\mathcal{R}' \longleftarrow \text{Child}(\mathcal{R}, \text{Dec}^t)$                  ▷ select $\text{Dec}$ sub-tree
13     $(G^t, sc^t) \longleftarrow \text{SampleCPDAG}(\mathcal{R}')$         ▷ a recursive call

14   sample $t' \sim P(t'; \{sc^t\}_{t=1}^{s})$ (see Equation 4)
15   $G_{\text{D}} \longleftarrow G^{t'}, sc_{\text{D}} \longleftarrow sc^{t'}$

16   $G \longleftarrow \cup_{i=1}^{k} G_{\text{A}i} \cup G_{\text{D}}$         ▷ recall that $G_{\text{D}}$ includes edges incoming from $G_{\text{A}i}$
17   $sc \longleftarrow sc_{\text{D}} + \sum_{i=1}^{K} sc_{\text{A}i}$        ▷ summation is used since the score is decomposable
18   **return** $(G, sc)$

---

# 4 Experiments

We use common networks[2] and datasets[3] to analyze B-RAI in three aspects: (1) computational efficiency compared to classic bootstrap, (2) model averaging, and (3) model selection. Experiments were performed using the Bayes net toolbox (Murphy, 2001). Conditional mutual information was used for CI testing, and BDeu with ESS $= 1$ for scoring.

## 4.1 GGT Efficiency

In the large sample limit, independent bootstrap samples will yield similar CI-test results. Thus, all the paths in $\mathcal{T}$ will represent the same single CPDAG. Since RAI is proved to learn the true underlying graph in the large sample limit (Yehezkel & Lerner, 2009), this single CPDAG will also be the true underlying graph. On the other hand, we expect that for very small sample sizes, each path in $\mathcal{T}$ will be unique. In Figure 3-left, we apply B-RAI, with $s = 3$, for different sample sizes (50–500) and count the number of unique CPDAGs in $\mathcal{T}$. As expected, the number of unique CPDAGs increases as the sample size decreases.

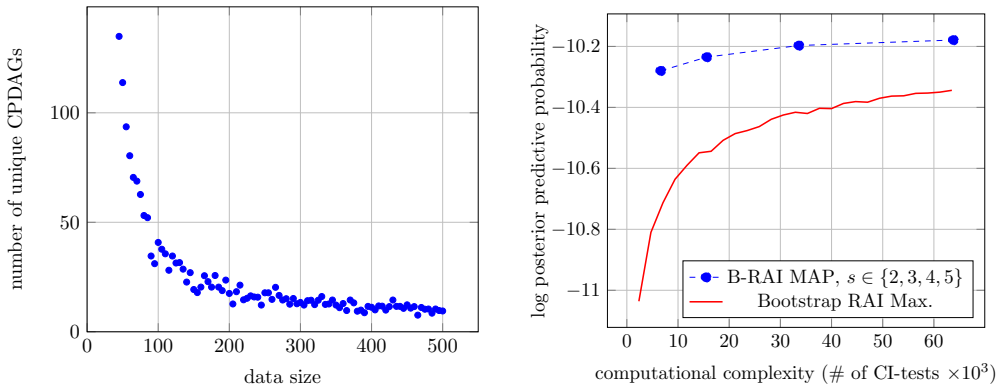

Figure 3: Left: Number of unique CPDAGs in a GGT with $s = 3$ as a function of data size (averaged over 10 different Alarm datasets). Right: predictive log-liklihood as a function of computational complexity (number of CI-tests). The MAP estimation from B-RAI, with $s \in \{2, 3, 4, 5\}$, is compared to selecting the highest scoring CPDAG from $l$ classic bootstrap samples (averaged over 1000 trials).

Next, we compare the computational complexity (number of CI-tests) of B-RAI to classic non-parametric bootstrap over RAI. For learning, we use 10 independent Alarm datasets, each having 500-samples. For calculating posterior predictive probability, we use 10 different Alarm datasets, each having 5000 samples. We learn B-RAI using four different values of $s$, $\{2, 3, 4, 5\}$, resulting in four different GGTs, $\{\mathcal{T}^2, \ldots, \mathcal{T}^5\}$. We record the number CI-tests that are performed when learning each GGT. Next, a CPDAG having the highest score is found in each GGT and the predictive log-likelihood is calculated. Similarly, for classic bootstrap, we sample $l$ datasets with replacement, learn $l$ different CPDAGs using RAI, and record the number of CI-tests. The different values of $l$ that we tested are $\{1, 2, \ldots, 27\}$, where the number of CI-tests required by 27 independent runs of RAI is similar to that of B-RAI with $s = 5$. From the $l$ resulting CPDAGs, we select the one having the highest score and calculate the predictive log-likelihood. This experiment is repeated 1000 times. Average results are reported in Figure 3-right. Note that the four points on the B-RAI curve represent $\mathcal{T}^2, \ldots, \mathcal{T}^5$, where $\mathcal{T}^2$ requires the fewest CI-test and $\mathcal{T}^5$ the highest. This demonstrates the efficiency of recursively applying bootstrap, relying on reliable results of the calling recursive function (lower $n$), compared to classic bootstrap.

## 4.2  Model Averaging

We compare B-RAI to the following algorithms: (1) an exact method (Chen & Tian, 2014) that finds the $k$ CPDAGs having the highest scores—$k$-best, (2) an MCMC algorithm (Eaton & Murphy, 2007) that uses an optimal proposal distribution—DP-MCMC, and (3) non-parametric bootstrap applied to an algorithm that was shown scale well for domains having hundreds of variables (Tsamardinos et al., 2006)—BS-MMHC. We use four common networks: Asia, Cancer, Earthquake, and Survey; and sampled 500 data samples from each network. We then repeat the experiment for three different values of $k \in \{5, 10, 15\}$. It is important to note that the $k$-best algorithm produces optimal results. Moreover, we sample 10,000 CPDAGs using the MCMC-DP algorithm providing near optimal results. However, these algorithms are impractical for domain with $n > 25$, and are used as optimal baselines.

The posterior probabilities of three types of structural features, $f$, are evaluated: edge,$f = X \rightarrow Y$, Markov blanket, $f = X \in \mathrm{MB}(Y)$, and path, $f = X \rightsquigarrow Y$. From the $k$-best algorithm we calculate the $k$ CPDAGs having the highest scores (an optimal solution); from the samples of MCMC-DP and BS-MMHC, we select the $k$ CPDAGs having the highest scores; and from the B-RAI tree we select the $k$ routs leading to the highest scoring CPDAGs. Next, for each CPDAG (for all algorithms) we enumerate all the DAGs (recall that a CPDAG is a family of Markov equivalent DAGs), resulting in a set of DAGs, $\mathcal{G}$. Finally, we calculate the posterior probability of structural features,

$$P(f|\mathcal{D}) \approx \frac{\sum_{G \in \mathcal{G}} f(G)P(G, \mathcal{D})}{\sum_{G \in \mathcal{G}} P(G, \mathcal{D})}, \tag{5}$$

where $f(G) = 1$ if the feature exists in the graph, and $f(G) = 0$ otherwise. In Figure 4 we report area under ROC curve for each of the features for different $k$ values, and different datasets. True-positive and false-positive values are calculated after thresholding $P(f|\mathcal{D})$ at different values, resulting in an ROC curve. It is evident that, B-RAI provides competitive results to the optimal $k$-best algorithm.

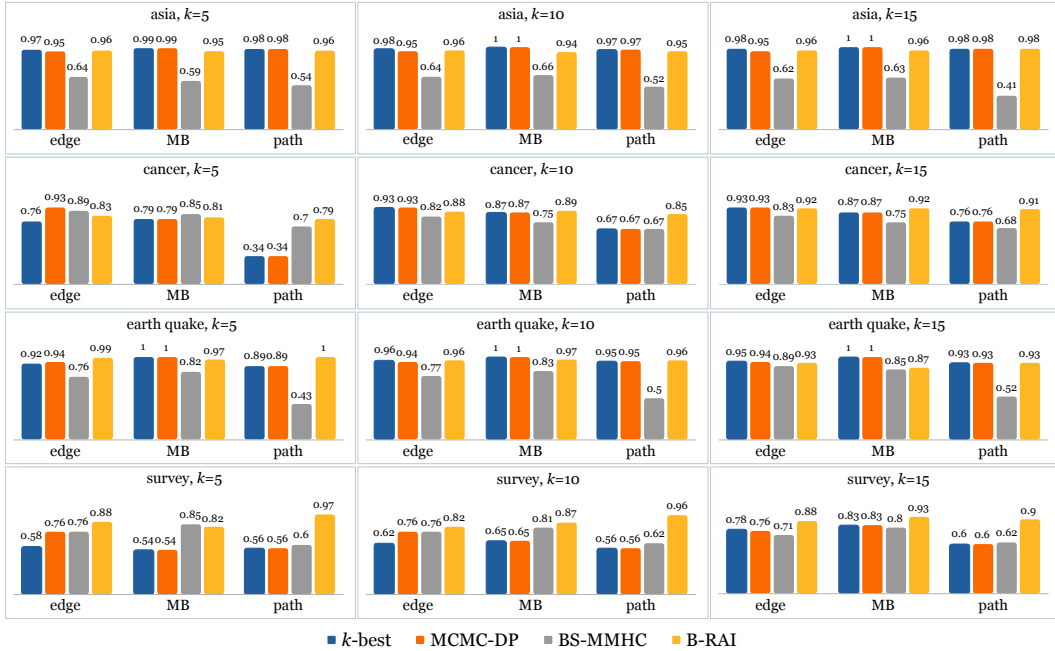

Figure 4: Area under ROC curve of three different structural features: a directed edge, Markov blanket (MB), and path. Each column of plots represents a different $k$ value ($k \in \{5, 10, 15\}$), and each row of plots represents a different dataset. Each bar-color represents a different method. The accuracy of B-RAI is on par with that of the exact $k$-best and MCMC-DP methods, and is better than the approximate bootstrap-MMHC method.

## 4.3 Model Selection

In this experiment, we examine the applicability of B-RAI in large domains having up to hundreds of variables. Since, optimal methods are intractable in these domains, we compare MAP estimation of B-RAI to three algorithms, RAI (Yehezkel & Lerner, 2009), MMHC (Tsamardinos et al., 2006), and classic bootstrap applied to MMHC, BS-MMHC. Both, RAI and MMHC, were previously reported to achieve state-of-the-art estimation in large domain. For BS-MMHC, 1000 CPDAGs were learned from bootstrap sampled datasets and the CPDAG having the highest score was selected.

We use eight, publicly available, databases and networks, commonly used for model selection (Tsamardinos et al., 2006; Yehezkel & Lerner, 2009). Each of the eight databases consists of 10 datasets, each having 500 samples, for training, and 10 datasets, each having 5000 samples, for calculating the posterior predictive probability. Thus, for each of the 8 databases, we repeat our experiments 10 times. Results are provided in Table 1. The longest running time of B-RAI (implemented in Matlab) was recorded for the Link dataset (724 nodes): $\sim 2$ hours. It is evident that B-RAI learns structures that have significantly higher posterior predictive probabilities (as well as scores of the training datasets; not reported). We also measured the structural hamming distance (SHD) to the true structure. For all datasets we found B-RAI to have the lowest percentage of missing edges, nearly $0\%$ of extra edges, and lowest direction errors. The smallest improvement of B-RAI compared to BS-MMHC was for Munin: $5\%$ fewer missing edges and $15\%$ fewer direction error.

Table 1: Log probabilities of MAP models

| Dataset | Nodes | RAI | MMHC | BS-MMHC 1000 Max. | B-RAI MAP $s = 3$ |
|---|---|---|---|---|---|
| Child | 20 | -68861 ($\pm$110) | -73290 ($\pm$60) | -72125 ($\pm$26) | **-65671** ($\pm$79) |
| Insurance | 27 | -71296 ($\pm$115) | -89670 ($\pm$118) | -85915 ($\pm$88) | **-70634** ($\pm$99) |
| Mildew | 35 | -288677 ($\pm$1323) | -296375 ($\pm$119) | -296815 ($\pm$270) | **-279686** ($\pm$1028) |
| Alarm | 37 | -52198 ($\pm$117) | -86190 ($\pm$220) | -80645 ($\pm$84) | **-51173.5** ($\pm$127) |
| Barley | 48 | -340317 ($\pm$389) | -380790 ($\pm$414) | -380305 ($\pm$380) | **-339057** ($\pm$804) |
| Hailfinder | 56 | -291632.5 ($\pm$259) | -308125 ($\pm$33) | -306930 ($\pm$95) | **-289074** ($\pm$120) |
| Munin | 189 | -447481 ($\pm$1148) | -455290 ($\pm$270) | -442860 ($\pm$255) | **-436309** ($\pm$593) |
| Link | 724 | -1857751 ($\pm$887) | -1907700 ($\pm$432) | -1863546 ($\pm$320) | **-1772132** ($\pm$659) |

## 5 Conclusions

We proposed a method that covers both model averaging and model selection in the same framework. The B-RAI algorithm recursively constructs a tree of CPDAGs, $\mathcal{T}$. Each of these CPDAGs was split into autonomous sets and bootstrap was applied recursively to each set independently. In general, CI-tests suffer from the curse-of-dimensionality. However, in B-RAI, higher order CI-tests are performed in deeper recursive calls, and therefor inherently benefit from more bootstrap samples. Moreover, computational efficiency is gained by re-using stable lower order CI test. Sampling CPDAGs from this tree, as well as finding a MAP model, is efficient. Moreover, the number of unique CPDAGs that are encoded within the learned tree is determined automatically.

In the large sample limit, independent bootstrap samples yield similar CI-test results. Thus, all the paths in $\mathcal{T}$ represent the same single CPDAG—the true underlying graph. On the other hand, we found that for a small sample size (small training set), each path in $\mathcal{T}$ represents a unique CPDAG. Thus, B-RAI has a virtue of inherently identifying the number of unique CPDAGs required to capture the distribution. It follows that for small samples sizes a larger $s$ (number of bootstrap splits) is required in order to capture a large number of CPDAGs in $\mathcal{T}$, whereas for relatively large sample sizes, a small $s$ is sufficient. This alleviates the computational cost.

We empirically demonstrate that while B-RAI has an accuracy that is comparable to optimal (exact) methods on small domains, it is also scalable to large domains, having hundreds of variables, for which exact methods are impractical. In these large domains, B-RAI provides the highest scoring CPDAGs and most reliable structural features on all the tested benchmarks.

## Footnotes

[2]www.bnlearn.com/bnrepository/

[3]www.dsl-lab.org/supplements/mmhc_paper/mmhc_index.html

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
