[Reviews · NeurIPS 2018]

Reviewer 1



This work expands on the algorithm RAI by Yehezkel and Lerner for constraint-based structure learning of Bayesian networks. The RAI algorithm constructs a tree of CPDAGs. Each node of the tree splits the variables into subsets (one descendant and K ancestral subsets) by using conditional independence (CI) tests of order n. Then, the node growths K+1 subnodes by recursively calling the RAI algorithm on the K+1 subsets of variables and n+1 as CI test order. The submission proposes the B-RAI algorithm that leverages bootstrap to allow the algorithm to output a set of highly likely CPDAG rather than the MAP one. Bootstrap is not naively leveraged. Instead, it is integrated in the recursive call of the algorithm. B-RAI constructs a tree similar to RAI except that B-RAI is recursively called S times on every subset, each time on a bootstrap sample of the dataset. This multiplies the number of subnode by S. The resulting structure can then be used to sample "boostrapped" CPDAG at a reduced cost. The resulting scheme is evaluated experimentally. It is compared to a naive bootstrap RAI for efficiency, to state-of-the-art model averaging schemes for structural features scoring on small data sets and to state-of-the-art structure learning scheme for data likelihood on large data sets. This paper has several strong points. - The experiments are very convincing. They show the advantage of the approach over a naive bootstrap and favourably compare B-RAI to RAI and state-of-art-approaches on two relevant problems on standard benchmarks. - This suggests this method could become a new reference approach. - The paper is mostly relatively clear and well written. On the other hand, the details of the algorithm proposed are not clearly explained in the paper. The key ideas are there but the paper is not self-contained. Reimplementing the algorithm cannot be done without at least reading the paper by Yehezkel and Lerner. I would suggest at least clarifying the following points in a supplementary material or in the main text. -- "Each group is independently partitioned", so would it be correct to say that, in Figure 1, n=1, "X^(1)_A1" refers to a different set of variables in each of the 4 partitions drawn? This was not clear to me. -- What are exogenous nodes? -- l 136-137: "First, the nodes having the lowest topological order are grouped into XD" --> Do you mean the nodes having the highest indexes in the topological order? How many nodes should be selected? All nodes without descendants? A set percentage of the nodes? How large are ancestors sets? Overall, I think this paper is interesting and worthy of publication, but might need to be clarified a bit. minor comments: - missing whitespace after comma in l115 - l 141: "therefore an sub-tree can further constructed independently" -------------- Thank you for the clarifications and the answers to my comments.

Reviewer 2



The paper addresses the problem of Bayesian network structure learning by providing an algorithm for sampling CPDAGs (the equivalent class of DAGs). The basic idea is to modify a constraint-based structure learning algorithm RAI by employing recursive bootstrap. It shows empirically that the proposed recursive bootstrap performs better than direct bootstrap over RAI. I think the paper is a useful contribution to the literature on Bayesian network structure learning, though not groundbreaking. It provides a method for dealing with the main weakness of constraint-based learning, that they are sensitive to errors in the independence tests, by a novel way of combining bootstrap with constraint-based learning. Overall the paper is clearly written. I don’t see any technical issues. The experimental evaluations are adequate, though it would be interesting to compare the proposed algorithm with those latest MCMC algorithms (e.g. Niinimaki & Koivisto 2013, Su & Borsuk 2016). I wonder what the complexity is of constructing a GGT (algorithm 1) and sampling a CPDAG (algorithm 2). Updates after author feedback: Thanks for the complexity analysis.

Reviewer 3



This manuscript proposes B-RAI, an extension of the Recursive Autonomy Identification (RAI) algorithm of Yehezkel & Lerner (JMLR, 2009), which is itself a constraint-based algorithm for Bayesian network structure learning. The idea of B-RAI is to use bootstrapping within each each RAI-substep in order to create a space of possible CDPAG (BN equivalence classes) solutions, for each of which a score is computed. So B-RAI belongs actually to the class of hybrids between constraint-based and score-based structure learning. The proposed approach has the advantage to cover both model averaging and model selection in the same framework. In summary, this is interesting work that could be useful to other researchers. The studied problem is important, and the results of benchmark studies look promising. The manuscript is generally written well. I have a few comments/questions, though: [1] The difference of B-RAI to RAI could be stated more explicitly early on (end of intro). Within Section 3, it is not easy to see where description of the old method ends and where the novel content starts. I had to look in the JMLR paper from 2009 to clarify some aspects. [2] The article employs BDeu as scoring function, but does not fully specify the prior: What is the chosen equivalent sample size (ESS)? This information is essential for reproducing the results. [3] If I understand correctly, the algorithm does not sample CPDAGs from the (full) posterior, but samples only CPDAGs that survive initial independence tests (and are thus included in the graph generating tree) according to their posterior probability. If this is so, it could be stated more explicitly, especially in the abstract. [4] For the model selection part, I miss a comparison with the globally optimal (CP)-DAG according to BDeu (with same ESS, see comment [2]) for the problem sizes (Child, Insurance, Mildew, Alarm, perhaps also Barley) where it can be computed with modern solvers (e.g. GOBNILP, CPBayes, URLearn). [5] For the model selection I also miss an evaluation of the solution's quality in terms of structural similarity to the ground-truth (measured by structural Hamming distance, for instance). It might be a more informative criterion than the log probability of MAP models, which is pretty hard to interpret (is a difference small or large?). [6] Very minor stylistic comments regarding the reference list: Please be consistent in using either full conference names or abbreviations (one "IJCAI" appears), and use of initial caps in paper/journal/conference titles. "Bayesian" instead of "bayesian", etc. [7] Another presentation issue: In Fig. 4, showing both the barplots and the numbers is a bit redundant. If the precise numbers are really important, a plain table might be a better way to present the results. Otherwise one could add a y-axis just to the leftmost plots (which then pertains to the entire row). ################## After rebuttal: Thank you for the detailed response, in particular clarifying [5]. Perhaps it's possible to add some result pertaining [4] to the final version [if completed until then], although it is not that critical, just a nice-to-have.